# An Intelligent Vision System for Detecting Defects in Micro-Armatures for Smartphones

**Jiange Liu**, **Tao Feng, Xia Fang, Sisi Huang and Jie Wang** *

School of Manufacturing Science and Engineering, Sichuan University, Chengdu 610065, Sichuan, China; liujiange666@163.com (J.L.); 15528248895@163.com (T.F.); FangXia1991@163.com (X.F.); hssylyn@163.com (S.H.)
* Correspondence: wangjie@scu.edu.cn; Tel.: +86-138-0801-5321

**Abstract:** Automatic vision inspection technology shows a high potential for quality inspection, and has drawn great interest in micro-armature manufacturing. Given that the inspection process is highly influenced by the lack of real standardization and efficiency performed with the human eye, thus, it is necessary to develop an automatic defect detection process. In this work, an elaborated vision system for the defect inspection of micro-armatures used in smartphones was developed. It consists of two parts, the front-end module and the deep convolution neural networks (DCNNs) module, which are responsible for different areas. The front-end module runs first and the DCNNs module will not run if the output of the front-end module is negative. To verify the application of this system, an apparatus consisting of an objective table, control panel, and a camera connected to a Personal Computer (PC) was used to simulate an industrial position of production. The results indicate that the developed vision system is capable of defect detection of micro-armatures.

**Keywords:** micro-armature; defect detection; convolutional neural networks; computer vision

## 1. Introduction

With the rapid expansion of network applications, computer vision technology has been successfully applied to the quality inspection of industrial production [1–6], including glass products [1], fabrics [2,3], steel surfaces [4], bearing rollers [5], and casting surfaces [6]. The inspection of these mentioned examples needs a matching algorithm to extract image features based on the actual defect situation. For glass products for packaging and domestic use, an intelligent system based on the classical computer vision technology was proposed for the automatic inspection of two types of defects [1]. For fabric quality control, an unsupervised learning-based automated approach to verify and localize fabric defects was proposed [2]. This approach was realized by using a convolutional denoising autoencoder network at multiple Gaussian pyramid levels to reconstruct image patches and synthesize detection results from the corresponding resolution channels. A two-fold procedure was proposed to extract powerful features [3]. First, the sample class was determined based on its background texture information, then the image was divided into 49 blocks to figure out which images contain defective regions. For the defects of steel surfaces, an inspection system with a dual lighting structure was proposed to distinguish uneven defects and color changes by surface noise [4]. In a previous study [5], a multi-task convolutional neural network applied to recognize defects was raised. Although there are many detection systems based on computer vision technology to solve product defects, few studies have focused on the defect inspection of micro-armatures. For the inspection of surface aluminum, a vision based approach and neural network techniques in surface defects inspection and categorization are proposed. The new vision inspection system, image processing algorithm, and learning system based on artificial neural networks (ANNs) were successfully implemented to inspect surface aluminum die casting defects [6].

Recently, deep convolutional neural networks (DCNNs) have been proved to be important methods in visual detection. However, some classical methods should still be considered. Literature [6] mentioned above is an example. For instance, in order to get better parameters, a suitable smart manufacturing strategy for real industrial conditions was proposed. The results of this dataset showed that the Adaboost ensembles provided the highest accuracy and were more easily optimized than ANNs [7]. Obviously, for now, there are some limitations with using classical methods alone or using DCNNs directly. For classical methods, for example, they usually come with high complexity of programming and less tolerance to data variability. With regard to the DCNNs, a large number of samples are needed for network training and the iterative optimization of parameters is partly a black-box operation [8].

In the industrial applications, part defects in the samples can be easily identified with a picture by the classical computer vision method. Therefore, we do not need to feed the whole picture into the network for discrimination, which can reduce the number of features to be identified. As a result, the network is easier to converge. Therefore, by combining the classical image recognition method and DCNNs, the landing speed of deep learning technology in industrial applications can be effectively improved.

In this work, an intelligent detection system that combined the classical computer vision method and DCNNs was designed to automatically detect the quality defects of micro-motor armatures. Firstly, the quality, excluding the region of copper wire crossing (ROC), is decided based on the classical computer vision technology. If the first result is positive, the ROC will be extracted and sent to the DCNNs for identification. If the result is still positive, the image is a defect-free sample, otherwise the whole image is labeled defective. In the experiments, this system works very fast and presents a high hit rate, which can bring practical benefits for industrial applications.

## 2. Related Works and Foundations

In motor armatures prepared for smartphones, the key component in the process of inter-conversion between mechanical energy and electrical energy is usually very small. During the service of the product, the poor performances of low-quality armatures will significantly affect the comfort of users. Furthermore, the magnetic field may change due to the poor quality of the armature, which can deteriorate the mechanical properties of the product.

Currently, many micro-motor armatures are manually placed under the microscope by the operator to adjust the armature position through the observation of the staff, according to the experience to achieve defect detection, which shows various disadvantages, such as time-consuming processes [9–11] and the lack of real standardization. Therefore, there is an urgent need to bring a related defect inspection system into the production process of micro-armatures.

As previously stated, currently the defect detection method for micro-motor armatures is achieved by transferring the armature to the inspection area after soldering, and then the armature defect is inspected by staff with microscopes. This detection method is not only expensive, but also inefficient and fluctuates with the flow of employees. To address these problems, the main aim of this work is, therefore, to design a new set of armature positioning and imaging devices, as well as the matching discriminating procedures. The armature and apparatus are shown in Figures 1 and 2, respectively.

Although the armature has three commutator terminals, a stepper motor was used to turn the armature. Thus, only one camera is needed to get the picture of each commutator terminal. We used the plane-array camera with 1.3 million pixels and the telecentric lenses with 0.66 mm depth of field and 110 mm working distance to capture the sample images. Fiber-optic sensor model E32DC200B4 and fiber optic amplifier model FX-101 were used to get the position of the armature.

Proper illumination can ensure the high quality of the image. We set up two area light sources, and a ring light source. The ring light source was arranged in front of the armature, and two area light sources were arranged on the left and right of the armature.

The software system was programmed with Python. The detection algorithm was developed by OpenCV and Tensorflow deep learning platform.

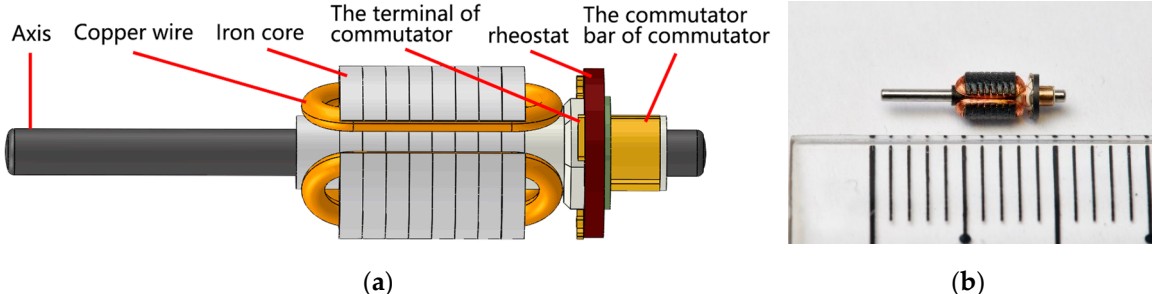

**Figure 1.** (**a**) The model of armature and (**b**) the real sample.

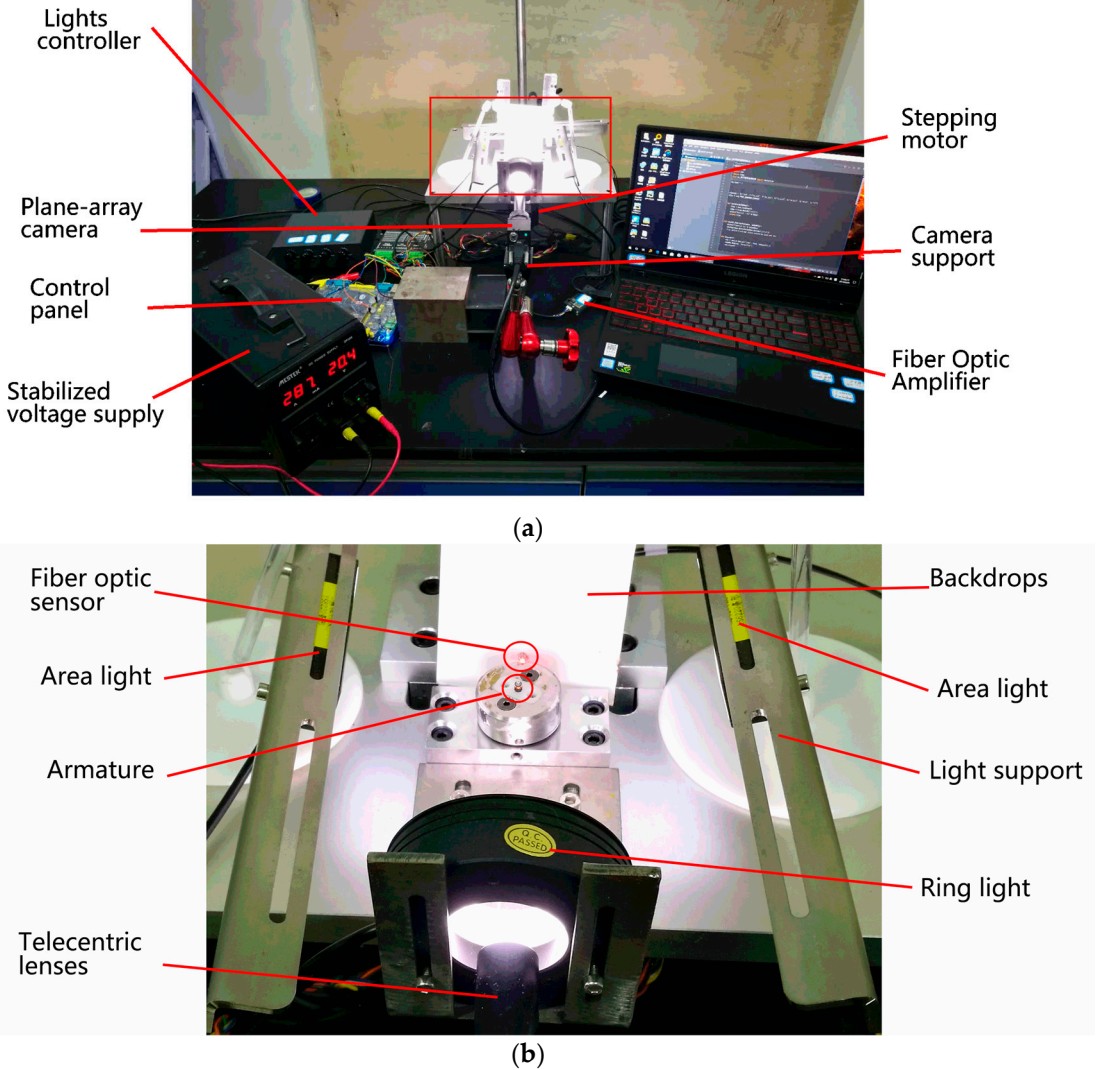

**Figure 2.** (**a**) Apparatus for the proposed inspection system. (**b**) shows the details in the red box of (**a**).

## 3. Methodology

Figure 3 provides an overview of the processing workflow, which displays the component of our proposed vision system. The system can be roughly divided into two parts, i.e., the classical computer vision method and DCNNs. Initially, the theoretical background for the model is depicted in Sections 3.1 and 3.2. Then, the database processed in this work is briefly described in Section 3.3,

while the details of the algorithm are discussed in Section 3.4. Finally, the experimental details are shown in Section 3.5.

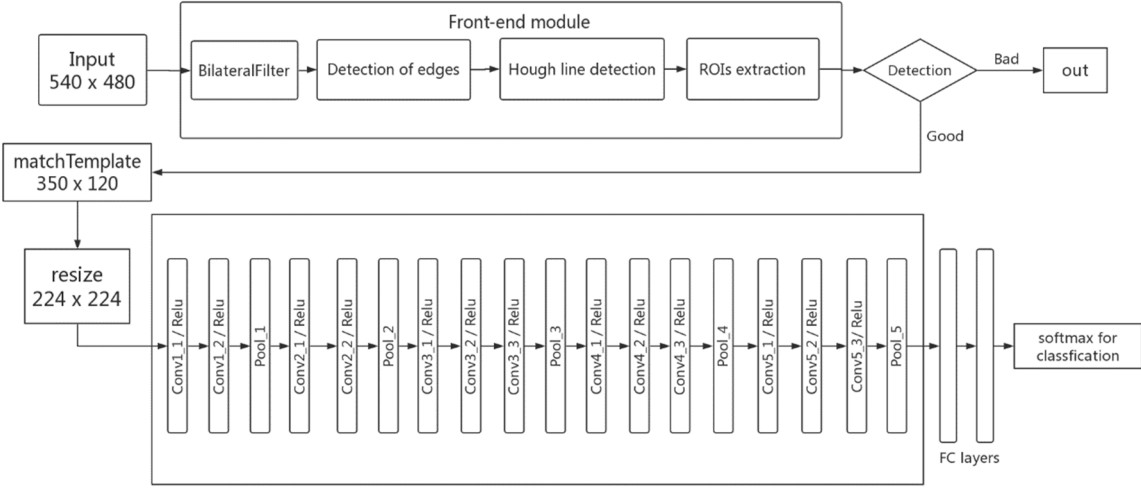

**Figure 3.** Architecture of our proposed intelligent vision system. The front-end module part takes the original images as inputs and the DCNNs part takes the extracted ROC as input if needed.

### 3.1. Front-End Module

#### 3.1.1. Filters

Bilateral filtering is a kind of non-linear filter [12,13], which represents the intensity of a pixel by weighted average of the brightness values of the surrounding pixels. The parameters of bilateral filtering take the Euclidean distance and color distance of the pixels in the range into account. Thus, the boundary is preserved, while the noise is reduced. Equation (1) gives the specific operation of bilateral filtering:

$$I(m) = \frac{\sum_{n \in \Omega} K_n H_n V_n}{\alpha} \tag{1}$$

where $I$ is the denoising image, $m$ is a pixel in the image, $n$ is an adjacent pixel with value $V_n$ in the $m$ neighborhood range $\Omega$, and $K_n$ and $H_n$ are, respectively, the space and gray weighting coefficient; $\alpha$ is the normalized constant, which represents the product of the spatial weighting coefficient and the gray weighting coefficient.

#### 3.1.2. Edge Detection

The purpose of edge detection is to find the pixels with sharp changes in brightness for an image, that is, a discontinuity in the intensity of grayscale. If the edges in the image can be accurately measured and positioned, it means that the actual object can be positioned and measured [14,15]. Many edge detection operations are based on the gradient and direction of brightness. The gradient and direction of an image g at point (x, y) can be separately defined by Equation (2) and Equation (3):

$$\nabla g(x, y) = \left[\frac{\partial g}{\partial x}, \frac{\partial g}{\partial y}\right] \tag{2}$$

$$\|\nabla g(x, y)\| = \sqrt{\left(\frac{\partial g(x, y)}{\partial x}\right)^2 + \left(\frac{\partial g(x, y)}{\partial y}\right)^2} \tag{3}$$

where $\partial g / \partial x$ and $\partial g / \partial y$ are the discrete partial derivatives in the x and y directions, respectively. With the appropriate threshold t, the edges can be detected.

### 3.1.3. Detecting of Lines

The Hough transform, proposed by Hough [16], is a method widely used in image processing and computer vision for detecting parameterized shapes. The simplest application is straight line detection, the main principle of which is firstly to convert the line detection problem in the image space to the point detection problem in the parameter space, then complete the line detection task by finding the peak in the parameter space. That is, if using Equation (4) to represent a line in the image space, the line is equal to the point $(\theta, r)$ in Hough space.

$$r = x * \cos\theta + y * \sin\theta \tag{4}$$

where $r$ is the distance between the point $(x, y)$ and the origin in image space, and $\theta$ is the angle between $r$ and the positive direction of x in image space. Note that some authors, such as Ballard [17] and Davies [18], have proposed enhancements to the method.

### 3.2. Deep Convolutional Neural Networks (DCNNs)

Recently, DCNNs has shown great detection power in computer vision, which has been widely used in various applications, e.g., classification [19], image segmentation [20], object tracking [21], and so on. A classic DCNNs architecture consists of several layers of convolutional, activation, and pooling layers, followed by fully connected layers at the end. A simplified version of DCNNs can be composed of the following five parts:

1.  Input: The input of DCNNs is usually a batch of 3-channel color image matrices with fixed size, which depends on the network structure you are using.
2.  Conv: The convolutional layers perform feature extraction and feature mapping by employing a set of fixed-size filters sliding on local receptive fields after receiving feature maps. The filter sizes are usually odd, such as $3 \times 3$ or $5 \times 5$. The weight-sharing scheme is applied in the convolution operations.
3.  Activation: Since convolution is a linear operation, it is necessary to use the activation layer to nonlinearly map the output of the convolutional layer, thereby increasing the expression ability of the model. Nowadays, rectified linear units (ReLU) have become the most widely used activation function because they can effectively prevent the gradient from disappearing and accelerate the convergence speed during the training process [16]. The mathematical transformation between each input value x and its output y can be formulated as

$$y = \begin{cases} x, & x > 0 \\ 0, & x \le 0 \end{cases} \tag{5}$$

4.  Pool: The pooling layer performs a form of non-linear down-sampling to compress the input feature map, which makes the feature map smaller in spatial size and reduces the complexity of the network computation. The most common pooling operation is max pooling, which outputs the max value from the neighborhood of the input feature map.
5.  FC: Each node of the fully connected layer is connected to all nodes of the previous layer to combine the features extracted from the front. The last fully connected layer generates the output of the overall network, the dimensions of which are the same as the dimensions of the input label, then achieving the classification with the transformation of the softmax classifier:

$$p(x_i) = \frac{exp(x_i)}{\sum_{i=1}^{k} exp(x_i)} \tag{6}$$

where $x_i$ indicates the value of the *ith* dimension computed by the last layer and $p(x_i)$ indicates the probability of the corresponding label.

### 3.3. The Dataset

In this paper, our dataset is collected by the device showed in Figure 2. We evaluate our method on this dataset, shown in Figure 4, which contains image samples of six representative defect types with a size $540 \times 480$ pixel. Further, we roughly divided the picture into two parts, respectively the ROC and the remaining region.

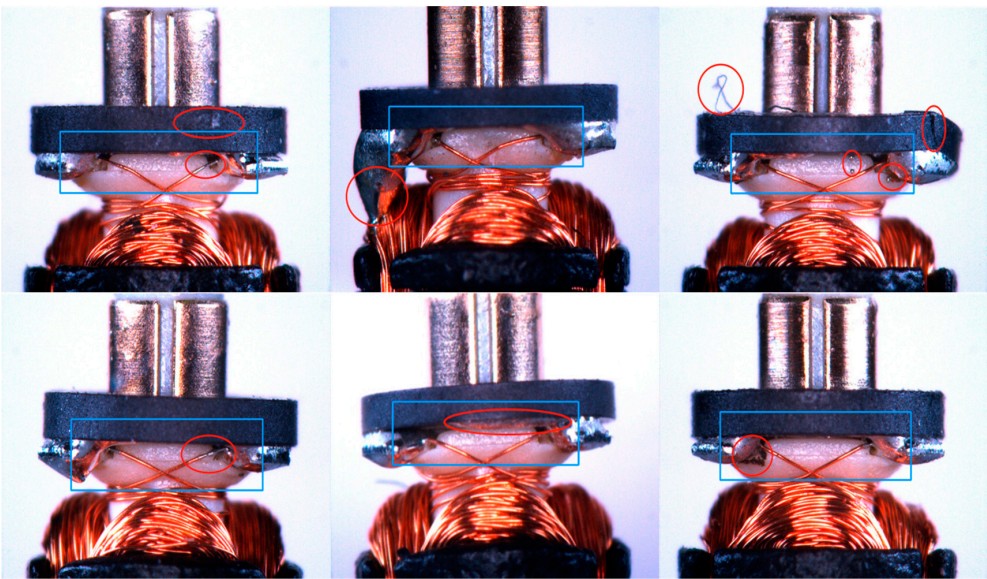

**Figure 4.** Original image samples taken by the device showed is Figure 2. The defective regions are marked out by a surrounding red ellipse; the ROC is marked by a blue rectangle.

For the ROC required to train the neural network, our dataset contains 5106 positive samples and 3322 negative samples; some of the samples are shown in Figure 5. As some pictures are prone to controversy in classification, we first assigned several groups of skilled employees to manually classify them, and then synthesized the classification results of several groups to finally determine the classification label.

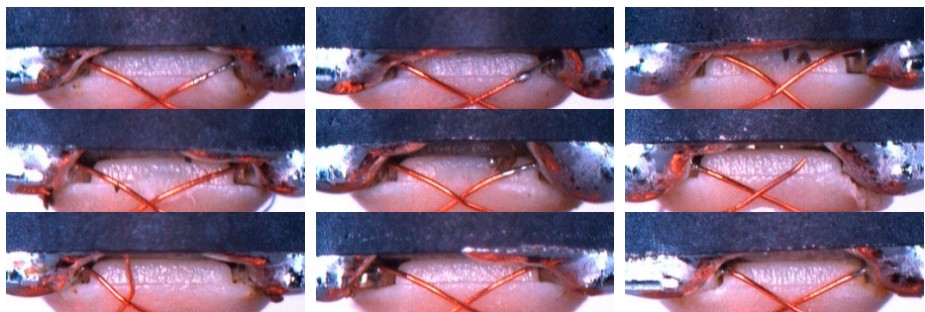

**Figure 5.** The ROC clipped from the original picture makes the proportion of defective parts larger, thus making the network easier to train and optimize.

### 3.4. Model Implementation

Considering the distribution of defective areas, as shown in Figure 4, the most complex defects are concentrated in the ROC (the region marked by a blue rectangle), and only this block is difficult to identify by the front-end module. Therefore, the implementation of our model is two-staged. We design a joint detection architecture, which contains two major parts: the front-end module and the DCNNs part. The front-end module was designed to detect these defects, including the resistance,

the tin package, and the cilia. The DCNNs detection part was developed to only identify whether the ROC contains defects or not. If the front-end module has determined that the armature contains defects, the DCNNs part will not be executed.

As shown in Figure 3, the front-end module can be roughly divided into five steps: filter, detection of edges, Hough transformation, regions of interests (ROIs) extraction, and identification. When the armature is placed on the workpiece table, the optical fiber sensor will conduct the armature to the initial position to make the sample face the camera. The picture at this point is shown in Figure 4. After the initial position correction is completed, the discrimination system starts to work.

The input of the system is a color image of the armature. In the front-end module, the RGB color image was firstly converted to a grayscale one by averaging the RGB channel. At this time, the effect of a canny edge detector used directly is shown in Figure 6. Note that Figure 6 shows a positive sample.

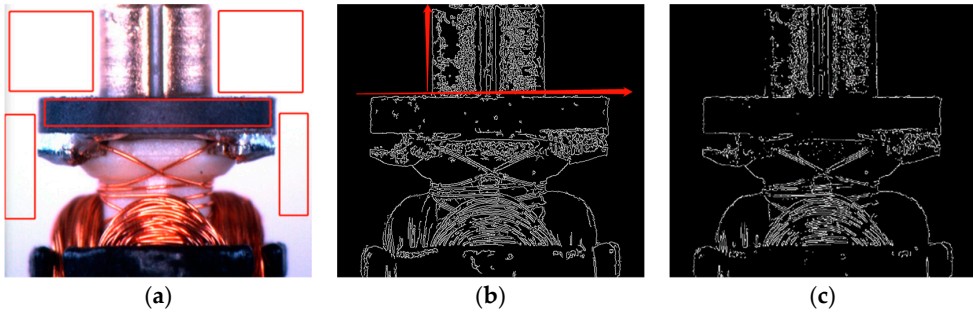

|    (a)    |    (b)    |    (c)    |

**Figure 6.** The original image and direct edge detection results. The ROIs have been marked by red rectangles (**a**). The two ideal baselines are marked by red lines (**b**). The noise is not reduced dramatically with the increase of the threshold, while the baseline disappears rapidly (**c**).

According to the above description, the areas we want to detect are the ones in the red checkboxes shown in Figure 6 in the front-end module. Combined with the edge detection results described by Figure 6, it is easily found that there are two obvious lines in the figure. A large number of pictures have proved that the two lines are stable, while both the horizontal line and vertical line are expected to be used as the reference line to locate other areas. However, the experimental result is different from the intuitive prediction, which may result from the inaccurate position of the lines detected by Hough transform. To improve the accuracy of detection, we increased the value of the double threshold to keep the edge and reduce the noise. However, the noise is not reduced dramatically with the increase of the threshold, while the baseline disappears rapidly. Thus, in our system, the bilateral filter is applied firstly to image noise reduction before edge detection. After the experimental verification of a large number of pictures, our canny operator achieved the best edge detection effect after the bilateral filter used the 60 mixVal and the 140 maxVal. The edge detection results are shown in Figure 7.

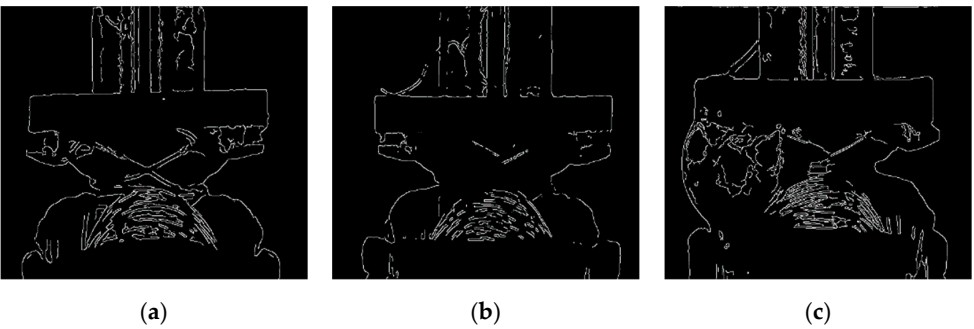

|    (a)    |    (b)    |    (c)    |

**Figure 7.** The edge detection results after using a bilateral filter. (**a**) shows that the picture is free of cilium and tin bags are too large defects; (**b**) reveals cilium defects. (**c**) represents the tin package is oversized.

In our dataset, the desired area can be located easily if the horizontal and vertical lines can be located stably. The experiment proved that the bilateral filter has no effect on the edge detection of the cilia and the tin package. The defects of the cilia and the package can be found directly by the edge detection result. For the three edge detection results shown in Figure 7, it can be found that the workpiece corresponding to (b) has cilium defects and the (c) tin package is oversized. For the rheostat region, we need to go back to the original image and cut it out, then use the new filter and threshold to carry out edge detection, and finally judge whether the rheostat is bad or not.

Although the poor resistance can be determined by edge detection according to the jump of color on the damaged location, it is easy to cause a lot of misjudgments if we still directly use the existence of the edge as the criterion, even with the new filter and threshold value. This is why firstly we convert the pixel edge to the pixel blocks by morphological transformation, secondly accumulate the area of the pixel blocks that exceed the threshold (although noise usually can be eliminated, some still converted into small pixel blocks), then set a threshold to output the result. The edge detection results of the resistance are shown in Figure 8.

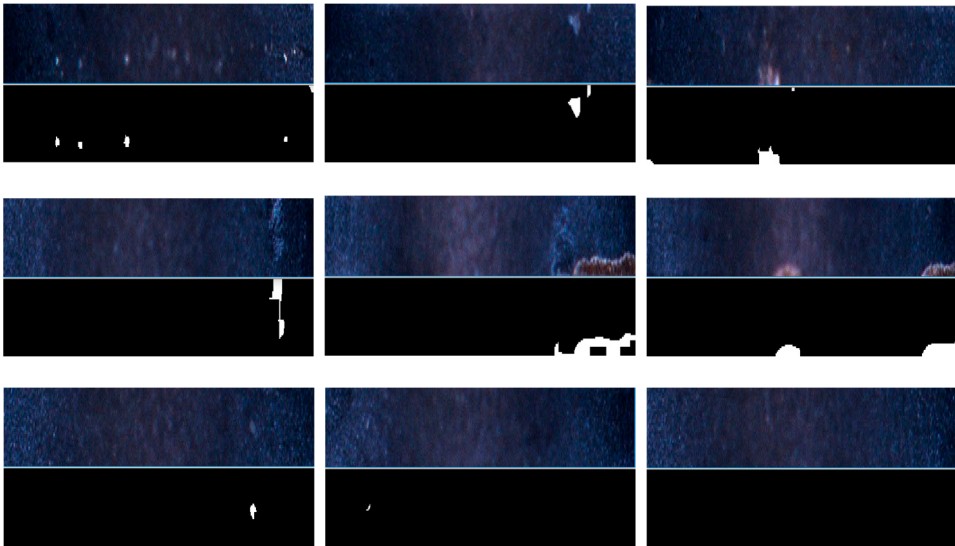

**Figure 8.** A total of nine images are shown. Each image is composed of two parts: the original image of resistance on the top and the edge detection image of resistance on the bottom. We will determine whether the resistance is damaged according to the white area in the edge detection diagram.

At this point, the inspection task of the front-end module has been completed. The system will directly judge whether the product is bad if any defective area is identified. Otherwise the template matching algorithm will be used to cut out the most difficult areas for discrimination and feed them into the trained neural network for detection. Thanks to the elimination of most interference, our DCNNs module did not need to be specially designed, while satisfactory results can be achieved directly using the classical classification network VGG_16 shown in Figure 3.

The front-end classical computer vision part works in conjunction with the DCNNs detection part for the purpose of defect detection. Given an image, the first part decides whether the external part is bad or not. If the external part is already defective, it is no longer necessary to enable the DCNNs part. If DCNNs is needed, it only needs to focus on a small part of the original picture.

*3.5. Experiments Details*

Our model is trained on one NVIDIA GTX1080 GPU with 8GB memory for roughly 10 h. Experiments are implemented based on the deep learning framework Tensorflow. The operating system is Windows 10.

The DCNNs network has proven its powerful ability in image detection, while the structure of DCNNs is very complex and deep, which is a kind of black box operation for us. At the beginning,

we directly fed the original image into to the VGG_16 network for classification [22]. However, overfitting occurred in the process of training the network. The highest accuracy of the test set was only 0.784, while the training accuracy is 0.998. Similar results also happened to Alex-net [16] and Resnet_50 [23]. We speculated that the reason for the failure is that with the multiple feature extraction, the network extracts more representative information and naturally loses details. However, in some cases, the differences between bad products and good products in our data lie in the small change; for some pictures, even people may be wrong. The original samples of data and ROI are shown in Figure 9. Therefore, we used the idea of SSD: Single Shot MultiBox Detector [24] for reference to integrate multiple scale features to make predictions based on VGG_16. To be specific, we concatenate conv5_3, conv4_3, and conv3_3 features, followed by a $1 \times 1$ convolution layer to form the final convolution feature representation. The method dose work and the accuracy was improved to 0.812, but it was still too low for our demand. In this case, we analyzed the data again and found that the three defects of armature winding cilia, resistance cracking, and tin package were relatively easy to be realized by classical computer vision technology, while only the small defect area where the two copper wires crossed was difficult to be distinguished by classical methods. We suspect that it may have achieved a higher accuracy rate if only such a small area was fed into the neural network. The experimental results show that our method is feasible. We obtained the difficult block with a size of $350 \times 120$ through template matching from each image to form the new training set [25]. Obviously, when a picture is switched left and right, the network needs to obtain the same discriminant result. Our workpiece does not switch up and down, nor does it tilt at an excessive angle. Therefore, in order to prevent the network from overfitting, we only adopt mirroring to augment data. In this way, the training set for the DCNNs now has 10,212 defect-free blocks and 6644 defective blocks.

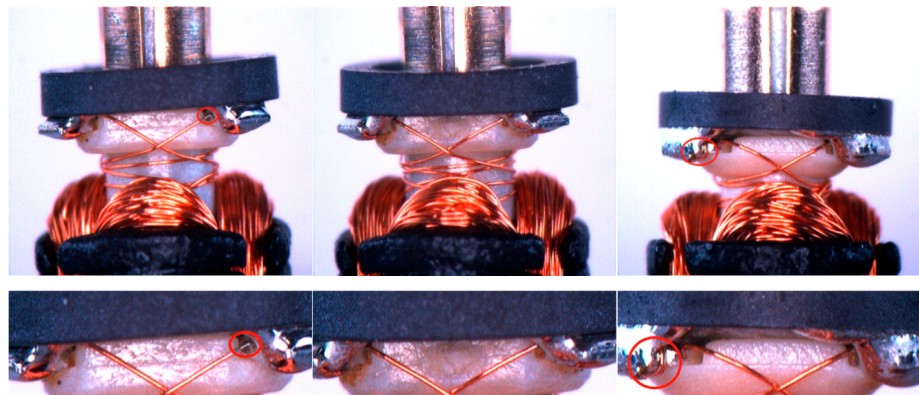

**Figure 9.** In some cases, the differences between bad products and good products in our data lie in the small changes, where even people may be wrong. However, at the same time, there are some areas in the picture that we do not care about and areas that are easy to identify, and the location of this area is relatively stable. If the whole picture is directly fed into the neural network, obviously we hope that the network can focus on the ROC region and give less consideration to other parts. In other words, we hope that the network can acquire stronger feature selection ability. However, according to our experimental results, the network does not have good feature selection ability. Therefore, we cut out the ROC from the original picture and only fed the ROC into the network, i.e., to help the network complete the process of feature selection through the traditional computer vision method.

Inspired by transfer learning [26–28], we used VGG16's convolution layers parameters trained on ImageNet as our initial convolution parameters. For FC layers, the weight parameters are initialized from a truncated random normal distribution subject to $N \sim (0, \frac{2}{n})$, where n denotes the number of connections between two layers. We selected the cross-entropy function as the loss function of our model. During the training process, the stochastic gradient descent with mini-batches of 16 samples

was applied to update the weight parameters. We set the base learning rate to 0.001, momentum to 0.9, and weight decay to 0.005.

## 4. Results

In our experiments, we used accuracy to evaluate the performance. The formula for calculating the accuracy is defined by Equation (7). To evaluate the performance of our method, we validated our module on our own dataset and achieved the final detection accuracy of 92.1%. The misclassified cases are listed in Table 1, which represents the confusion matrix, and the comparisons to directly feed the total image into network are listed in Table 2. It can be seen from Table 2 that the defective inspection can be achieved by directly feeding the total image into the network with accuracy of about 80%, whereas our two-stage module acquired a final score of over 92% in the classification task. We believe that this comparison result verifies our previous conjecture; that is, the network does not acquire good feature selection ability when the original image is directly fed into the neural network, resulting in low accuracy. The experiment shows that our model has the ability to distinguish between defect-free and defective images in our dataset and achieve a higher accuracy than others, which proves the effectiveness of our two-stage module.

$$Accuracy = \frac{TP + TN}{TP + TN + FP + FN} \tag{7}$$

where *TP* represents the number of positive sample that are judged to be positive samples, *FP* represents the number of negative samples that are judged to be positive samples, *FN* represents the number of positive samples that are judged to be negative samples, and *TN* represents the number of negative samples that are judged to be negative samples.

**Table 1.** Confusion matrix.

|  | **Predicted Positive** | **Predicted Negative** |
|---|---|---|
| **True Positive** | 2785 (*TP*) | 207 (*FN*) |
| **True Negative** | 187 (*FP*) | 1809 (*TN*) |

**Table 2.** The detection results of our model and the comparison to only DCNNs. "B345" represents the combination of "conv3_3", "conv4_3", and "conv5_3", "*FM*" represents our front-end module, "VGG-16-ROC" means that the neural network input is only ROC.

| Method | Accuracy (%) |
|---|---|
| VGG-16 | 78.4 |
| AlexNet | 78.2 |
| ResNet-50 | 78.3 |
| VGG-16+B345 | 81.2 |
| FM | 95.8 |
| VGG-16-ROC | 93.6 |
| VGG-16+FM | 92.1 |

## 5. Conclusions

In industrial applications, workpiece images in many cases are similar to our dataset due to automation and standardization. ANNs and complex algorithms are in use today in artificial vision, and the use is in some cases very useful. For example, some defects of a sample can be easily detected by classical computer vision technology, while the combination of deep learning techniques, and traditional computer vision techniques should be considered. Our experimental results provide a reference and demonstration for the cooperation of DCNNs and complex algorithms.

In this study, a system based on classical computer vision and deep learning was proposed to detect the micro-armature defects. In our dataset, the rough location of defects is relatively stable,

and partial defect detection can be easily achieved by classical computer vision technology. In order to improve the accuracy of our system, for the defective parts that are easy to be identified, we will achieve the defection work by classical computer vision technology, and then only feed the complex parts into the neural network if needed. This is similar to L1 regularization, which can reduce part parameters of the network to 0 and achieve the effect of parameter sparsity, for which we use the front-end module to make the neural network have similar ability. We conducted many experiments, the results of which proved that our method was superior in accuracy, and met the requirements of industrial manufacturing.

In general, the main contributions of this work are as follows:

1. According to the workpiece, the supporting fixture, optical positioning system, and lighting system are designed, and the defect detection algorithm matching our hardware is designed.
2. Through experiments, it is proven that the combination of DCNNs and complex algorithms is very useful in some cases, providing a reference and demonstration for the application of computer vision technology in industrial detection.
3. In our work, we combine traditional computer vision technology and DCNNs to achieve the task, so as to improve the detection accuracy. This two-stage idea could also be considered for use in deep learning techniques, i.e., a two-stage approach similar to Faster R-CNN [29]. This two-stage idea is worth considering as we focus more on accuracy.

The difficult in our experiments is that some of the ROCs are very prone to controversy in classification. More data processing methods will be explored to solve this problem. Note that we are not talking about data augmentation. In the future, we will continue to optimize the front-end module and network structure to improve the accuracy.

**Author Contributions:** The authors J.L. and T.F. contributed equally to this work; project administration, J.W.; resources, X.F. and S.H.

**Funding:** This presented work was supported by Sichuan Science and Technology Program (No. 2019YFG0359).

**Conflicts of Interest:** The authors declare no conflict of interest.

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
