# Peer review of "An Intelligent Vision System for Detecting Defects in Micro-Armatures for Smartphones"

_applsci, doi:10.3390/app9112185_

Round 1

Reviewer 1 Report

This manuscript proposes an apparatus and method for detecting defects on micro-armature. In the proposed algorithm, the region of copper wire crossing is extracted as a region of interest by applying edge detection and Hough line detection algorithm, and a deep neural network based on the architecture of VGG was used to classify defect and defect-free images. Although novelty of the proposed algorithm is relatively low compared to other articles in Applied Science, apparatus and experiment setting are expected to receive attention of readers in the field of industries.

To improve the quality and completeness of the manuscript, I suggest following two comments: At first, the measure for evaluating accuracy was not explained (in Table 1, line 280), and evaluation method should be clearly described by using recall rate (sensitivity), precision and/or f1-score. The detailed explanation and effectiveness of the data augmentation (in line 257) will be helpful to readers. Experiments to compare the result without data augmentation can show the effectiveness the data augmentation.

Author Response

Thank you for your letter and for the reviewers’ comments concerning our manuscript entitled “An Intelligent Vision System for Detecting Defects in Micro-Armatures for Smartphone”. Those comments are all valuable and very helpful for revising and improving our paper, as well as the important guiding significance to our researches.We would like to take this opportunity to thank you for making the detailed comments showing the tremendous effort in reading our paper. The main corrections in the paper and the responds to your comments are as flowing:

(1) Point 1: the measure for evaluating accuracy was not explained (in Table 1, line 280), and evaluation method should be clearly described by using recall rate (sensitivity), precision and/or f1-score.

[Answer] The measurement for accuracy evaluation in our experiments has been supplemented in Results (line 310), and one confusion matrix was added.

(2) The detailed explanation and effectiveness of the data augmentation (in line 257) will be helpful to readers. Experiments to compare the result without data augmentation can show the effectiveness the data augmentation.

[Answer] The detailed explanation of the data augmentation has been provided (in line 291). In our experiments, the trick of data augmentation was directly used at the beginning after analysed the dataset, however we apologized that the result without data augmentation was not compared.

Reviewer 2 Report

The paper presents a machine vision system for the defect inspection of micro-armature used in smartphones.

Some suggestions:

Related work of micro-armature inspection should be given in Related works and foundation section. It is only said that nowadays is inspected by operators with a microscope or other tools. How is it this procedure?. Which are the other tools?.

Further detail on the devices included in the computer vision system (camera, optics, lighting system) in Section 2.

Figure 1 and 2 should be improved. Figure 1, label the armature elements. Figure 2, label the elements of the proposed vision system.

Result section should be improved. Only one table and a paragraph is not enough to validate the proposal. Results could be improved with a confusion matrix and a repeatability and reproducibility analysis.

The conclusion is not supported by the results

Author Response

Dear Reviewers:

Thank you for your letter and the comments concerning our manuscript entitled “An Intelligent Vision System for Detecting Defects in Micro-Armatures for Smartphone”. Those comments are all valuable and very helpful for revising and improving our paper, as well as the important guiding significance to our researches. We would like to take this opportunity to thank you for making the detailed comments showing the tremendous effort in reading our paper.  The main corrections in the paper and the responds to your comments are as flowing:

(1) Related work of micro-armature inspection should be given in Related works and foundation section. It is only said that nowadays is inspected by operators with a microscope or other tools. How is it this procedure? Which are the other tools?

[Answer] We have moved the related work of micro-armature to Related works (section2, from line 29 to 79), and supplemented the description of inspection procedure (in line 84). We are very sorry for our incorrect writing ‘or other tools’, there is no other tools, only microscope.

(2) Further detail on the devices included in the computer vision system (camera, optics, lighting system) in Section 2.

[Answer] We have added the further detail on the devices in section 2 (in line 96)

(3) Figure 1 and 2 should be improved. Figure 1, label the armature elements. Figure 2, label the elements of the proposed vision system.

[Answer] We have improved the Figs. 1 and 2.

(4) Result section should be improved. Only one table and a paragraph is not enough to validate the proposal. Results could be improved with a confusion matrix and a repeatability and reproducibility analysis.

[Answer] We have supplemented a confusion matrix and do more description to our experiments in result section.

(5) The conclusion is not supported by the results

[Answer] We have refined our result and discussion to a more detailed description.

Round 2

Reviewer 1 Report

The revision shows substantial improvements by addressing all the concerns of the reviewer. I have not further comments and I believe that it reaches the borderline for publication.

Author Response

Dear riviewers:

Thanks very much for your kind work and consideration for our paper. On behalf of my co-authors, we would like to express our great appreciation to editor and reviewers.

Thank you and best regards.

Yours sincerely,

Reviewer 2 Report

The authors have improved the paper and all the recommendations have been included

Author Response

Dear reviewers:

Thanks very much for your kind work and consideration for our paper. On behalf of my co-authors, we would like to express our great appreciation to your.

Thank you and best regards.

Yours sincerely,